# Adults' Perceptions on Adolescent Attitudes towards Pregnancy and Abortion in Maputo and Quelimane Cities, Mozambique: An Exploratory Qualitative Study

Mónica Frederico [1,*], Carlos Arnaldo [2], Rehana Capurchande [3], Peter Decat [1] and Kristien Michielsen [1]

[1] International Centre for Reproductive Health, Department of Public Health and Primary Care, Faculty of Medicine and Health Sciences, Ghent University, C. Heymanslaan 10 UZ, 9000 Ghent, Belgium
[2] Centro de Estudos Africanos, Eduardo Mondlane University, Department of Development and Gender Studies, Maputo P.O. Box 1993, Mozambique
[3] Department of Sociology, Eduardo Mondlane University, Campus Universitário Principal, Maputo P.O. Box 257, Mozambique
[*] Correspondence: monica.frederico@ugent.be

**Abstract: Background**: In Mozambique, 76% of adolescents have been pregnant before the age of 20 years. Thus, this study explores adults' perceptions on adolescent attitudes towards pregnancy and abortion in Maputo and Quelimane cities. **Methods**: A qualitative study was conducted in Maputo and Quelimane, with four focus group discussions. A cross-sectional household survey was used to select adult women participants. Data were analysed applying a thematic analysis approach. **Results**: Intrapersonal, interpersonal, cultural, and environmental factors influence pregnancy and abortion decision making among adolescents. Generational conflicts reduce the importance paid to traditional knowledge transfer, contraceptive beliefs, denial of paternity, lack of parental support, and procreation value were found to influence abortion decision making and early pregnancy among adolescents. **Conclusions**: There is a need to improve relationships between adults and adolescents to reduce girls' vulnerability to early pregnancy, as well as empowering adolescents in order to negotiate safe sex, reducing unwanted pregnancy and induced abortion.

**Keywords:** pregnancy; abortion; decision making; law; adolescents; Maputo; Quelimane; Mozambique

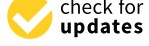



## 1. Introduction

Of the estimated 10.2 million unintended pregnancies occurring each year among women aged 15–19 years in developing countries, 3.3 million result in unplanned births, and 5.6 million in abortions, of which 3.9 million are unsafe, and 1.2 million are miscarriages (Darroch et al. 2016). In sub-Saharan Africa, data from 2008 to 2012 reveal a high adolescent pregnancy rate, ranging from 121 (Ethiopia) to 187 (Burkina Faso) per 1000 adolescents, with equally high abortion estimates, varying from 11 per 1000 pregnant adolescents in Ethiopia, over 21 in Malawi, to 38 per 1000 adolescents aged 15–19 years in Kenya (Sedgh et al. 2015).

In Mozambique, pregnancy and abortion are major issues affecting young women. Data from 2015 Malaria Indicator Survey show that 76% of female adolescents aged 19 years had been pregnant. This survey also showed that 23.5% of them reported that the pregnancy was unplanned (Ministério da Saude and Instituto Nacional de Estatística 2018), which increases the risk of pregnancy termination. Further, the general prevalence of modern contraceptive methods in Mozambique is low, especially among female adolescents aged 15–19 years (only 15.7%) (Ministério da Saude and Instituto Nacional de Estatística 2018), which increases the risk for unintended pregnancies and subsequently induced abortion. For example, Pizzol et al. (2018) found that in Beira (Mozambique) 3021 (36%) out of

8290 adolescent girls and young women aged 15–25 years who accessed the Six Friendly Services for Adolescents clinics in 2014 have been pregnant, with most of them being in the 15–19 age band (59%). This scenario happens in a context where teenage pregnancy and abortion, especially unsafe abortion, can cause serious consequences for women.

Currently, induced abortion is accepted in Mozambique. Before the 1980s the criminal code inherited from the colonial administration was restrictive regarding pregnancy termination (Hardy et al. 1997; Agadjanian 1998). However, since the introduction of a rule by the Ministry of Health in the 1980s, authorizing hospital abortions in specific cases (i.e., women who got pregnant while using an intrauterine device (IUD) or if the woman's health was in danger), the interpretation of this decree has become more flexible, and abortion regarded as "quasi legal" (Hardy et al. 1997; Agadjanian 1998). In 2014, a liberal abortion law was adopted in Mozambique (Assembleia da República 2014) allowing women to have an induced abortion before up to 12 weeks of pregnancy (Ministério da Saúde 2017). This reduced the contribution of abortion on maternal mortality registered in Maputo Central Hospital, from 11% in the 90s to 8.6% in 2015–2018 (Machungo et al. 2019).

In Sub-Saharan Africa, female adolescents' decision making regarding sexual and reproductive health (SRH) issues, especially on contraceptive use, is influenced by social and cultural norms (Bain et al. 2021; Ochako et al. 2015). These norms are transmitted by parents as custodians of the tradition. In this process mothers play an important role (Shams et al. 2017) in the education of their daughters for a safe and healthy transition to adulthood (Sooki et al. 2016; Izugbara 2015).

They teach their daughters healthy behaviours and shape their discourse on all issues (Shams et al. 2017; Wamoyi et al. 2015). Based on these norms of control and regulation of the adolescent girls' sexuality, it is not socially acceptable for a girl being sexually active before marriage (Buller and Schulte 2018), since virginity represents good conduct (Bhana 2016), and being sexually chaste and modest in dressing characterizes a good girl (Lenzi et al. 2018). So, parents/guardians believe that limiting sexual health education for adolescent girls is a way to protect them from risky sexual behaviour (Shams et al. 2017). This description may reflect the fact that mothers lack knowledge about sexual issues as well as skills for effective communication with young people. They also feel embarrassed to discuss sexual issues with their daughters, fear the arrogance as well as fearing encouraging them in sexual experimentation, since it is believed that discussing reproductive health with adolescents motivates their curiosity (Shams et al. 2017; Motsomi et al. 2016). Additionally, adults perceive adolescents as too young to understand sexual issues, in addition to the lack of a conducive environment for open discussions of sexual and reproductive health matters (Motsomi et al. 2016). What has just been described is reinforced by the study of Usonwu et al. (2021), which pointed out that a lack of parental self-efficacy, as well as cultural and religious norms, create an uncomfortable environment which leaves peers, media, teachers, and siblings as important and sometimes preferred sources of sexual health information. Similarly, Mcharo et al. (2021) pointed out that young people reported that it was difficult to discuss sexual and reproductive health matters with their parent/guardian. Despite all these aspects, girls are socialised and prepared as future wives and mothers (Wamoyi et al. 2015; Igras et al. 2014).

While parents try to preserve their cultural and social value, the demand for pregnancy-related health services by adolescent girls and young women is increasing. Considering this role played by adult women the in social and cultural education of adolescent women, associated with the fact that, in Mozambique, existing studies about early pregnancy and induced abortion are generally based on the perception of adolescents who experienced these events, this paper aims to fill the gap on female adults' perceptions regarding adolescents' sexual and reproductive health, in particular, about pregnancy and induced abortion.

The objective of this study is to explore adults' perceptions on adolescent attitudes towards pregnancy and abortion in Maputo and Quelimane cities. This study was based on the argument that female adults, such as mothers and other guardians, are the first social entity with whom adolescents and young women interact and learn about sexual

and reproductive health, influencing and shaping their behaviour regarding sexual issues. Considering that the interaction of the two groups is not isolated from the community and the society as a whole, we used the ecological model, which is a comprehensive framework (Bronfenbrenner 1994) that allows us to structurally screen for causes at the intrapersonal, interpersonal, cultural, and environmental levels, as well as to explore possible interventions to address these issues.

Thus, the study is a contribution to a better understanding of the decision-making processes among adolescent women in Mozambique. It is also a contribution to the existing literature by identifying those who have power in the pregnancy prevention and abortion decision-making process, and by understanding when, with whom, and how this power is exercised.

## 2. Methodology

### 2.1. Study Site

This study is a part of broader research on pregnancy during adolescence and youth in Maputo and Quelimane cities in Mozambique, and it was conducted in urban districts in Quelimane (District No. 2 and District No. 4), and Maputo (KaMaxaquene and Ka-Mubukwana). These urban districts were selected due to the high proportion of pregnancy and induced abortion among women aged 15–24 that showed in the quantitative data also collected in the context of the broader study.

Maputo is the capital of Mozambique, and it is located in the south of the country, while Quelimane is the capital of the central province of Zambézia. These two cities are different regarding the kinship system of their family organizations as well as in the way they prepare their daughters and boys for the transition from childhood to adulthood. While Maputo has a patrilineal kinship whose descendance is traced following the father's lineage, Quelimane has a matrilineal one in which responsibility with children and descendance follows the mother's lineage. Yet in these communities, girls and boys are submitted to sexual initiation rituals in which they are instructed on how to live in the community, including preparation for marriage.

### 2.2. Data Collection

This paper is complementary to the previous article "Factors Influencing Abortion Decision-Making Processes among Young Women" (Frederico et al. 2018) that focused on the perceptions of young women themselves.

Data were collected through Focus Group Discussions (FGDs), between July and August 2017, as a part of a broader research on pregnancy during adolescence and youth in Maputo and Quelimane cities in Mozambique. The focus group discussions were semi-structured, following a guideline with specific topics that served as entrance for a given aspect, followed by probes. During the discussion, it was not important to follow the order in which the topics were outlined, but to ensure that all topics were addressed. Two researchers were involved in these FGDs, one as moderator (the principal researcher) and the other one as note-taker (assistant). The discussions were recorded with the permission of all participants who had been informed about all procedures. Adult women were asked about pregnancies in their communities, whether they were frequent, whether they were increasing compared to previous years, and about their perception regarding the causes of adolescent pregnancy and how the communities dealt with this issue. Participants were also asked about their position and perceptions regarding abortion by adolescents and were invited to share their opinion about the new status of abortion in Mozambique.

In total, four focus group discussions (two in Maputo and two in Quelimane) of nine to twelve participants were conducted. Potential participants were identified through the household survey in which respondents could express interest in contributing further to the study. The selection was purposive, and some of those that expressed interest were visited and invited to participate in the study, which focuses on adults' perception on adolescents' pregnancies and induced abortion decision making. The criterion used to

invite the participants was based on age (equal to and above 25 years), availability, and interest. These selection criteria included women, as members of the study areas, to share their views and perceptions about pregnancy occurring among young women in their communities, as well as to reflect on the new law on abortion. However, they were not selected because they themselves had experienced pregnancy/abortion in the past. The participants were asked to reflect on pregnancy trends in their community, comparing the years before the abortion regulation and after. The discussions were conducted in Portuguese and Xichangana (Maputo) and Echuwabu (Quelimane).

### 2.3. Data Analysis

The data were transcribed, translated, and analysed. The transcription was made by the principal researcher (MF) who also led the focus group discussion. The evaluation of the quality of the transcription was performed by another researcher (RC) involved in the study. The analysis was based on the ecological model and consisted of three steps: reading, coding, and analysis in NVivo version 11. A structured thematic analysis was used to make inferences and elicit key emerging themes from the text-based data (Bryman 2012). It started by grouping all responses according to questions relating to each of the topics discussed (pregnancy, abortion, and the new abortion law). This was followed by the coding process, which grouped causes into intrapersonal, interpersonal, cultural, and environmental. The coding tree was developed by the first author (MF) and then the codes and the classification were discussed among the co-authors. Interviews in local languages were translated into Portuguese, and then all transcriptions were translated into English.

### 2.4. Ethical Consideration

Ethical approval was obtained from the Institutional Committee of the Faculty of Medicine of Eduardo Mondlane University, the National Bioethical Committee for the health of the Ministry of Health (IRB00002657), and the ethical committee of the Ghent University Hospital (PA 2015/043). We also obtained an authorization from the Minister of Health and local authorities at the provincial and community levels. The participants gave their informed consent after the objectives and procedures had been explained to them. This study is complementary for the broad study which involved young women who had ever experienced an induced abortion and was developed to improve our perception regarding what is happening in the community. For this group of participants, there was no support in terms of health care and counselling. However, this was provided for the young women in the broader study. For example, in Quelimane, the researcher had to refer two girls to the health facility and NGOs providing counselling. One of them was psychologically affected after losing her mother due to rape. The second had contracted a sexually transmitted disease.

### 2.5. Limitation

A major limitation of this study is the fact that participants were only from two cities, which does not allow the findings to be generalized to other cities of Mozambique, even those of a similar culture and societal organization. This study reports only the point of view of female adults, which cannot reflect adolescent womens' perspectives, nor that of adolescent and adult men.

## 3. Results

### 3.1. Participants' Characteristics

Table 1 summarizes the characteristics of the FGDs participants. The participants were aged 25 to 62 years. Of all fourty-five participants, only six had completed secondary school, thirty-two had completed primary school, and seven had no education. Almost all participants were Christian and four were Muslim. Three participants were students, thirty-three were wage earners, and nine were unoccupied. Participants in Maputo were

mainly occupied in tertiary activities, and in Quelimane, the majority were in primary activities.

**Table 1.** Socio-demographic characteristics of the focus group participants.

| Categories | Maputo | | Quelimane | |
|---|---|---|---|---|
| | Group 1 | Group 2 | Group 1 | Group 2 |
| Number of participants | 12 | 12 | 9 | 12 |
| Age range in years | 25–59 | 36–62 | 29–57 | 25–60 |
| Education attainment | | | | |
| Without formal education | 0 | 5 | 0 | 2 |
| Primary school | 9 | 6 | 9 | 8 |
| Secondary school | 3 | 1 | 0 | 2 |
| Religion | | | | |
| Christian | 12 | 12 | 8 | 9 |
| Muslim | 0 | 0 | 1 | 3 |
| Occupation | | | | |
| Without occupation | 2 | 3 | 2 | 2 |
| Student | 3 | 0 | 0 | 0 |
| Vendor | 7 | 7 | 0 | 0 |
| Farmers | 0 | 1 | 7 | 10 |
| Receptionist | 0 | 1 | 0 | 0 |

*3.2. Adolescent Pregnancy*

This section presents female adults' perceptions of pregnancy regarding adolescent women. These perceptions were grouped based on the levels of the socio-ecological framework.

3.2.1. Intrapersonal Level

Early sexual activity was perceived as a cause of pregnancy. Adolescent women are blamed for engaging early in sexual activity that results in pregnancy, which is often unwanted. These circumstances are a risk factor for induced abortion.

*Even you will be upset to see [who is only] 12 years old [but is pregnant]. (Quelimane)*

*This is the reason we see girls of 12, 13, 15 years old pregnant because they initiate sexual activity before menstruation. (Maputo)*

To prevent early pregnancy, participants mentioned that adolescent women should focus on their studies to prevent them from engaging in sexually risky behaviour that may result in pregnancy. This will also assure a better future, since they will continue with their education process. However, the participants also understand that other adolescent women focus on school after experiencing pregnancy.

*If the girl gets pregnant, you must tell her: 'do not give up studies'. (Maputo)*

*There is a child that listens to you when you tell her: "attention, study! To reach your goal tomorrow". Others learned after having [pregnancy] and regret. (Quelimane)*

Adults also think that adolescent women should focus on religion to prevent early pregnancy.

*So we only [have to] give [them] many tasks, [such as] sending them to the church, and enrol them in the dominical school to avoid bad jogs [sexual activity]. (Maputo)*

*We have to start sending our daughters to the church early. Going to church she will grow hearing the word of God and she will not be interested in carnal matters. (Quelimane)*

From the data, it is clear that an intrapersonal factor such as engaging in first-time sex at an early age is a factor that influences early pregnancy. So, from the adult perception, focus on education and religion can contribute to delay sexual initiation.

### 3.2.2. Interpersonal Level

Female adults pointed out the lack of dialogue due to generational conflicts. Adult women claimed that adolescent women do not take their mothers' advice on how to behave because they're considered as old-fashioned ideas.

> *In our homes, we suffer because our granddaughters do not respect us. When we try to talk to them, they just say grandmother you are old-fashioned. (Maputo)*

> *Our daughters are now offending us. They do not listen; they do not want to know. So we just stay like this, asking "what are we going to do?" (Quelimane)*

As a consequence of this troubled intergenerational communication, adults are often kept ignorant about their adolescent daughter's first menstruation, only becoming knowledgeable about this when they find dirty clothes. Because of that, mothers do not have the opportunity to instruct their daughters according to their tradition.

> *If our daughters were growing up following the tradition, it would be good because it forces you to inform the aunt when the menstruation starts, and in turn, the aunt gives you the treatment according to the tradition. Currently, there is nothing we know about our daughters. (Maputo)*

> *The daughter does not tell us anything. The girl that you saw at my house did not tell us that the menstrual cycle had initiated. Who discovered it was her father and told me to ask her 'what is happening?'. (Quelimane)*

Participants indicated that friends and peers play a role in young people's lives, in the sense that they imitate the actions of one another to feel integrated into the group.

> *Even before they reach 10 or 11 years, they fall in the same situation [get pregnant] because their friends say: let's play. (Maputo)*

> *When you [as a mother] explain to her [the daughter], [you] do not play like that, take care! But because she saw her friends who acquired trousers, skirts, fancy hair, she starts doing the same [transactional sex] like her friends, to be able to acquire similar things, and then she will get pregnant. (Quelimane)*

As can be understood from the findings, there is intergenerational conflict between adults and young women on how to practice sexuality. This conflict contributes to the young women becoming influenced by friends and peers. They appear to uptake risky behaviour, such as the partner's refusal to use a condom. Improvements in intergenerational dialogue is seen as a way to reduce early pregnancy.

### 3.2.3. Cultural and Environmental Levels

At this level, causes of early pregnancy identified by adult women were sexuality education at school, menstruation cycle initiation, the means of knowledge transfer, time for sexuality conversation with adolescent women, and beliefs about contraceptive methods. Media and poverty were also mentioned.

The school was mentioned as a cause because it teaches reproduction-related issues. For some participants, the idea of adolescent women receiving reproduction lessons from school takes away the opportunity for mothers to introduce sexuality, menstruation, and pregnancy prevention talks with their daughters. However, for others, the information given at school helps them to introduce sensitive talks with adolescent people, as they are not prepared to have such conversations among themselves.

> *I asked her, did you start to menstruate? she said yes. When? and she said 4 months ago. Who gave you the instruction? She said I learned at school. You see, will she inform us this [menstruation], even though it is your obligation [to teach her], as a mother? They will not say anything because they learn everything outside the home and at school. (Maputo)*

*They learn at school. The teacher of Biology subject explains all these things openly. They have to explain that there is menstruation and that once menstruation is initiated, meeting a man [if you have sexual relation] you will get pregnant and this pregnancy is a risk. Also inform the children that they should deepen this issue with their mothers, to see if from there I can be able to open myself [and talk about it] because I want but I can't. (Quelimane)*

Regarding the early onset of menstruation, adult women perceive this phenomenon as the result of early sexual activity, also referred to as the cause of early pregnancy.

*But today, adolescent girls sleep [have sexual intercourse] with a man, before having their first menstruation, this is what causes menstruation. So they start to see [have menstruation] very early, at the age of 9 years old. (Maputo)*

*The girls now begin dating early before menstruation. This is why menstruation appears too early. When you dig a hole like this [a metaphor for sexual intercourse], you find water. If not, you do not find it, isn't it? (Quelimane)*

Adult women recognize the need for a conversation with adolescent women to be careful in their relationships and to use contraceptives consistently to prevent pregnancy. According to them, the conversation has to occur from the moment that the mothers perceive that their daughters have changed their behaviour and that they are sexually active.

*If you perceive that the girl started to date, you have to have an open conversation. You have to explain to her that she has to use a condom to prevent pregnancy and STD, because if she already started it is not possible to force her to give up because she wants. (Quelimane)*

*But if you suspect that this girl is having bad behaviour [dating], talk to her in a good way and show her the condom. Say, my daughter, when your time reaches, you have to be with this thing [condom] to avoid pregnancy and disease. (Maputo)*

FGDs showed that the transfer of knowledge regarding menstruation from mothers to daughters is surrounded by taboos and intimidation. There is no clear message that mothers transmit to their daughters to make them well-informed and, thus, to help avoid early pregnancy, either through prevention or through postponing sexual onset. Such taboos prevented some women from openly informing their daughters about menstruation. This resulted in the use of metaphors such as "you are injured" or "you are grown" to describe menstruation and the care that their daughters have to take to avoid pregnancy. Other participants explained that when menstruation starts, the first procedure is to ask the young woman what happened, followed by a threat and the accusation of having had early sexual intercourse.

*When the breasts grow earlier, the grandmother or aunt was sweeping the girl's breast to stop it. When the menstruation was initiating, the person was crying because she was thinking that she was injured. At that time, the adults were saying: indeed, you were cut off by a blade. Because of that, you cannot play with the boys because they will hurt you a lot. So you have to be a quiet girl. (Maputo)*

*When my mother understood I started to have menstruation She threatened me: you slept with a man, you see what came out? Tell me the truth. For my daughter, I also did the same, threatened her and she started crying and saying no mom, I did not do anything. (Quelimane)*

Beliefs were also expressed regarding the use of contraceptive methods by adolescents because of their effect on fertility. Despite this, some participants pointed out the need for using contraceptive methods for avoiding pregnancy, while others mentioned that there is no need to use contraception before experiencing pregnancy.

*I do not know if she can procreate. So how can I inform my daughter that [she] has to avoid pregnancy? How to prevent pregnancy before she experiences birth, before seeing something?" (Quelimane)*

*Me as a mother, I cannot take a girl of this age and insert an implant because we do not know what God put in her womb. When the time to have a baby reaches and she faces difficulties, she will accuse me of being responsible for her infertility because it was me who told her to use contraceptive methods. (Maputo)*

Regarding media, participants referred to the television as the vehicle through which adolescents learn about sexual intercourse.

*I do not know if it is because of television or conversations that they always have, that make them knowledgeable about these things (sexual intercourse). There is a day that my neighbour called me because there were small children having sex, do you see? (Maputo)*

*[In] soap opera [they see] man kissing a woman, man caressing a woman, sometimes pornographic movies. On television, they see how parents have sex. So, they keep in mind that there is nothing harmful, and they implement it. (Quelimane)*

From the quotations above, television is seen as being responsible for inciting adolescent people to have sex because it shows soap operas portraying romantic content, from which adolescent women can learn and be motivated to engage in sexual intercourse and relationships.

Poverty was also mentioned as an environmental cause that increases the risk of early pregnancy. In the context of the two study sites, adolescent women are reported to engage in sexual activity with a man who is economically stable and able to sustain their needs.

*The young women now date with a married man, who uses cars to deceive them, giving 2000 MZN [28.6 euro] and on the other day give her whatever. (Quelimane)*

*Now it is difficult! Some girls start to date due to poverty. (Maputo)*

The first quotation illustrates the current situation imposed by inequalities in economic conditions. It shows that those who are economically powerful use their condition to seduce adolescent women in order to have transactional sex. In such conditions, it is difficult for adolescent women to negotiate protected sex. The second quotation shows another perception of the strategy used by a partner to convince a girlfriend to have unsafe sex, a strategy that involves telling her that adults are not saying the truth and persuading her not to use a condom in sexual intercourse.

*Boys do not want to use the condom. They want meat by meat [unprotected sex], while the girl wants what she asked for [mobile phone, artificial hair]. What does she do? She will accept it. (Quelimane)*

*The man can say or ask: "is this [condom] for what? They [adults] are lying to us [that it is necessary to condom use]. Let's try [sex] without it [condom]". (Maputo)*

From the participants quotes, it is possible to understand that factors such as a lack of sexuality education at school, taboo, beliefs, misinformation, myths regarding menstruation cycle initiation, contraceptives methods, time for sexuality conversation, means of knowledge transfer, influence of television, poverty, and transactional and unprotected sex constitute the main cultural and environmental factors influencing early pregnancy.

### 3.3. Abortion Decision-Making

#### 3.3.1. Intrapersonal

Participants indicated the circumstances in which the pregnancy occurred as important for the decision to keep or terminate it. Pregnancy can occur at an early age when the adolescent is not prepared for motherhood. It can also occur while adolescent women lack financial resources. These circumstances lead to the perception that adolescent women terminate the pregnancy because they will have other children in the future. The decision

of pregnancy termination can also relate to the case of multiple partners and uncertainty about the father of the baby. Quotes below.

> *They [think they] will have another baby in the future. (Maputo)*

> *If a man says guys, we all are going to wash our dirty hands here; it means that the girl had 5 guys. Because you did not settle in one place. They refuse the responsibility for that pregnancy [and you decide to terminate it]. (Maputo)*

Early age, continuation of studies, lack of resources to take care of the baby, and multiple partners are interpersonal factors influencing the abortion decision making.

### 3.3.2. Interpersonal

Participants of both study sites mentioned the influence of peers or friends, refusal of paternity by the partner, and lack of support from the parents. However, mothers may have little influence since, in many cases, parents are not even informed about the decision-making process regarding abortion.

> *They deceive together. A friend says: my friend, this is nothing. Once I got pregnant and I went to a nurse and I gave her some money [and I induced abortion]. (Quelimane)*

> *The girls now agree with others. We do not know where they find pills. We only know that there is someone who knows pills and places where they can go and induce abortion. (Maputo)*

The partner's fear of responsibility for the pregnancy on the one hand, and the girl's fear of being rejected by the partner on the other hand, are also considered factors in pregnancy termination.

> *They agree with their lover to go to the hospital and induce abortion because he does not want to assume responsibility. (Maputo)*

> *Sometimes they tell the boyfriend about pregnancy and the boy says it is better to induce abortion. And she starts to think that if I continue with pregnancy I will lose the boyfriend, so I have to abort. (Quelimane)*

Findings also reveal that parents may not accept the pregnancy.

> *Other mothers say: you impregnate, who will assume it? You have to go to the man who impregnates you or you have to abort. (Maputo)*

> *Some parents decide to terminate the pregnancy because the boyfriend refused pregnancy responsibility. (Quelimane)*

The quotes illustrated that pressure from friends about postponing the pregnancy, rejection of paternity, lack of parental support, and fear of parents are factors at the interpersonal level that influence the abortion decision among young women.

### 3.3.3. Cultural and Environmental Causes

Regarding cultural and environmental causes, very little emerged from the findings. It is likely that this may be linked to perceptions and notions surrounding pregnancy. Most participants understood that all pregnancies should end at birth. This perception is possibly due to the importance and meaning that the child has, which can afford a way for adolescent women to move past their mistakes, as well as the symbolic importance of a child, seen as an enrichment of the family.

Regarding the perception of learning from their mistakes, the participants explained that having a baby while they are still young will force them to prevent another pregnancy, through suffering by looking after the baby, and going to school at the same time.

> *The better thing is to continue with the pregnancy, send her to school and she will see other girls with nothing. So she will feel bad. But this thing that the law says that if she wants to terminate, she can, it is not learning and also is not correct. (Maputo)*

*[Continuing pregnant] she will see, after facing some consequences, expenses of the baby, the responsibility, and uncomfortable [condition]. From that point, she will not want to have another pregnancy. (Quelimane)*

While the idea that keeping a pregnancy until birth is a lesson for adolescent women was common in both study sites, the perception that having a child is equal to family enrichment was mainly found in Quelimane. Every pregnancy is welcomed because the community highly values the family. Among the participants, there was also the perception that in Maputo, an unwanted pregnancy could result in induced abortion, or the new-born could be left alone anywhere, which means that the value given to a child is here considered less compared to Quelimane.

*The wealth they value is the children. They do not even look at age, 15 or 16. If she is pregnant, they do not try to terminate it. It is rare here in Zambézia to hear that a mother has left a child, as it happens in Maputo. The enrichment of the family is not the work. It is just to have children. (Quelimane)*

At cultural and environmental levels this study found different meanings and prestige regarding children as factors influencing the decision related to the outcome of pregnancy. In the two contexts of the study, being able to bear a child is connected to notions of masculinity and femininity.

*3.4. Perception of Female Adults Regarding the 2014 Abortion Law*

During the FGDs, the abortion law of 2014 was discussed. The result showed a lack of consensus regarding the status of abortion in Mozambique.

First, the non-acceptance of the new abortion law was related to the perception that the availability of legal abortion will stimulate unprotected sexual intercourse because adolescent women know that they have a chance to induce an abortion.

*This law is not good for girls, because they will impregnate sure that they will induce abortion. The law is for those who recognize and correct their failures. (Maputo)*

Second, adult women who do not accept abortion understand that abortions are sought by pregnant women who are rejected by the partner due to their promiscuity. This reveals stigma and judgment in the community.

*If the man does not accept the pregnancy, this is because it was the girl who told the man, "I want to date you", while she is also dating other boys. So you can't say, the law is the law. (Maputo)*

Third, adult women also do not accept induced abortion at a health facility because they think several contraceptive methods can help to prevent pregnancy.

*There are injections, pills, IUD [for prevention]. So, why they have to request an abortion? (Quelimane)*

The acceptance of the new abortion law is related to the perception that the permission of pregnancy termination at a health facility will help pregnant women who are very young avoid unsafe abortions. This demonstrates that the position regarding abortion services vary according to circumstances.

*I think this law is not wrong because there are little girls [pregnant] and the law will help them and avoid the use of traditional medicine or to buy pills anywhere and use it [without assistance]. (Quelimane)*

*We cannot force a girl to continue with pregnancy while she does not want. The baby will suffer because the girl who gave birth to him or her will [in future] say they forced me to give birth. (Maputo)*

For adult women, abortion is also acceptable if the woman is ill, or if the pregnancy occurred due to the failure of the contraceptive methods.

*To request abortion depends. If the woman is sick, then she can. (Maputo)*

> *Someone is using contraceptives and suddenly sees that she is pregnant and has 4, 5, 6 children. So this is the case to induce abortion. (Quelimane)*

Participants were also asked about their views on the new abortion law, and the findings reveal a diversity of perspectives. These diversities can be explained by how each participant understands abortion and pregnancy. For some, the law of abortion is incorrect because young women can prevent pregnancy using contraceptives that are free of charge and are available in health facilities. Abortion is also seen as being against procreation, and so it is stigmatized and judged. For others, the approval of the new law is important because it helps young women to postpone pregnancy.

## 4. Discussion

This paper provides female adults' perceptions of pregnancy and abortion decision making among adolescent and young women in Maputo and Quelimane cities in Mozambique. The female adults' perceptions allowed us to identify causes of pregnancy and abortion decision making among adolescents at the intrapersonal, interpersonal, and cultural-environmental levels. However, personal causes of abortion were mentioned only by participants of Maputo.

Thus, we structured the discussion around two topics: (1) intergenerational conflicts and reduced importance paid to traditional knowledge transference, and (2) beliefs about contraception and the importance of procreation in the Mozambican culture. In our interpretation, these messages constitute the key lessons from the data.

For example, intergenerational conflict and reduced importance paid by traditional knowledge emerged as one of the main lessons of this study because, during the data analysis, we noticed that the argument presented by the adult women to explain the occurrence of early pregnancy, as well as the decision to induce abortion, was the fact that young women do not listen to their advice because they (young women) consider that those advices are outdated. It is also understood that beliefs about contraception and the importance of procreation in the Mozambican culture constitute another lesson for this study, which is that adults, in their talks about adolescents and young women's sexual and reproductive health, do not consider the use of contraception as an option for pregnancy prevention. This thought is because of beliefs and the misconceptions embedded in community. Based on the argument of adult women, pregnancy prevention must occur after proving the capacity to bear a child to the family. Before this, there is no need to practice prevention because it is not known if the girl will be able to procreate, and talk of contraception is perceived as influencing negatively in this regard. The selected topics may be integrated into the interpersonal and cultural-environmental levels of factors, taking into account that they show how relationships are constructed and guided, especially between adults and young women. These topics were selected not only for being the most crucial from the findings, but also for being more critical and conflicting from the participants' points of view. It is not possible to explicitly indicate which level of factors gives a better explanation for, for example, intergenerational conflicts, which are seen as a combination of different levels of factors that contribute to this situation. However, in the following paragraphs we focus on each topic.

Intergenerational conflicts create a weak dialogue between adolescents and their parents which is a barrier to the transference of information about sexual and reproductive health between adults and adolescents. Indeed, while young women do not tend to seek information from older generations, they do consult other sources. Based on our earlier studies and literature review, we have identified peers or older siblings and radio, television, and other media (Mcharo et al. 2021; Igras et al. 2014) as the main sources where young women feel free to ask and share their doubt, without judgement, because they have showcase similar needs and perceptions of how to live their sexuality. In the Mozambican context, knowledge is passed from the adult to young people, since they are considered individuals with incipient knowledge when compared to adults' knowledge. According to this thinking, the best way to grow up is by listening to information from

adults. Young women that are too self-confident may be considered problematic and sometimes thought to lack respect (Wamoyi et al. 2015). This was also suggested by adult women who participated in this study.

These attitudes from adults leads adolescents to engage in sexual activity without information on how to prevent pregnancy. This finding is consistent with other studies in Sub-Saharan Africa, showing a weak dialogue about sexuality between parents and children (Yadeta et al. 2014; Nundwe 2012; Obono 2012; Yibrehu and Mbwele 2020) and conflicts between generations (Reynolds Whyte et al. 2008). In both study sites, mothers reported that they were accused by adolescents of having outdated ideas. This can be explained by differences in socialization, particularly considering that adolescents are currently exposed to the influences of the media. They are also exposed to changes in the culture, which is seen as contradictory to the traditional way of living experienced by the female adults' generation. This reveals the difficulties that adults face in guiding their daughters. Mothers insist on providing the education that they received when they were young in a period where media exposes young people to another culture. This is similar to the study of Reynolds Whyte et al., which showed the conflicts of generations expressed in the adults' insistence on saying that they had an honourable and admirable life in a morally superior era (Reynolds Whyte et al. 2008). Female adults were frequently recalling the comparison between the time they grew up, where they were not told directly that bleeding means menstruation, and how this information was not questioned. Currently, adolescents question the elder generation about "do and do not" instructions related to sexuality, while the elder generation continues to be reluctant to openly explain these issues. This can constitute as an influence for adolescent women to seek this information from other sources, for understanding biological processes causing changes in their sexuality, or for solving problems of unwanted pregnancies. The adult reluctance may be related to a lack of adequate knowledge about sexuality (Kamangu et al. 2017; Kajula et al. 2013; Tegegne et al. 2019; Simmonds et al. 2021), or because culturally it is a taboo (Kajula et al. 2013; Simmonds et al. 2021; Motsomi et al. 2016; Frederico et al. 2019). The weak dialogue can also be seen with other family members, such as godmothers/aunts (Kamangu et al. 2017), those who used to be key persons in counselling adolescent women about socially acceptable behaviour (Kamangu et al. 2017) as well as keeping communication lines open between mothers and daughters, since mothers usually feel uncomfortable discussing sexuality with their daughters (Shams et al. 2017; Kamangu et al. 2017; Motsomi et al. 2016). The perception of the lack of respect mentioned by most participants may also be linked to existing conflicting views as well as intragender power differences. Age and sex are two categories of hierarchy in the context of this study influencing gender relations and distributing differentially knowledge and power between adults and young women. When knowledge is shared by other group members, especially those who are still young, they are met with anger and even violence (Duby et al. 2022). As it has been found in other studies, there is a generational gap in how topics around sex are approached/perceived, and proscriptive socio-cultural guidelines around sexuality communication inhibit open communication; furthermore, in some communities, overt discussion of sexual matters is regarded as unacceptable, especially in front of adolescents and young women (Duby et al. 2022; Buller and Schulte 2018).

Adult women mentioned having been accused of having antique ideas by adolescents, which created generational conflicts. However, they do not see their role as influential in the generational conflict. The findings of Frederico et al. (2018) indicated that parents were one of the factors that influence abortion decision making among young women in these two study sites by threatening to kick them out of their home. This shows that the parents use power instead of dialogue to impose rules on young women. The difficult interaction between adult women and adolescents requires an intervention to improve the dialogue between the two groups, as was indicated in a pilot intervention in Maputo, Mozambique (Frederico et al. 2019).

Data analyses were illustrative regarding how the conversation about sexuality between mothers and daughters is not common in the Mozambican context, due to generational conflicts. This results in adolescent women experiencing the first menstruation without informing their mothers or guardians. The menstrual cycle initiation is perceived by adults as an important event in both study sites, where parents, especially female adults, are expected to be informed by their daughters when it occurs, in order to proceed with the respective traditional ritual. Culturally, the first menstruation is a moment in which parental power control is applied to make sure that reproduction takes place under controlled and institutionalized conditions, i.e., marriage (Arnfred 2011). It is also the moment in which young women are submitted to the traditional ritual of initiation, which serves as an opportunity to instruct and transmit local habits and beliefs according to the tradition. During this process, the transfer of knowledge related to the menstruation cycle, from mothers to daughters, is often accompanied by intimidation; use of metaphors such as "injury" to describe menstruation and referring to the cares that must be taken to avoid pregnancy, as well as misinformation about menstruation, are prevalent. This ambiguity leads to adolescent women experiencing fear when it comes to informing their parents, not only about menstruation but also other events occurring during adolescence (Nundwe 2012; Obono 2012; Yibrehu and Mbwele 2020; Maina et al. 2020). When schools give information about menstruation to adolescent women, mothers, grandmothers, and aunts lose the opportunity to instruct them about cultural norms which they consider as their obligation to do. In study sites, menstruation is understood not only as a biological but also as a social and cultural issue that needs to be oriented based on cultural norms and transmitted by grandmothers, aunts, or ancients (Nundwe 2012; Obono 2012; Reynolds Whyte et al. 2008). Given the difficulties in talking about sexuality at a younger age, adolescents are ignorant about puberty and developmental changes (Craig and Richter-Strudom 1983). Parental communication starts late, after puberty or after sexual onset, and is often judgmental about premarital sex and induced abortion (Yibrehu and Mbwele 2020; Maina et al. 2020). Consequently, it is not surprising that in the context of adolescent abortion, the preferred person to talk about it with is another young person, one who is living the same experiences.

Focusing on school and religion was considered a good way to prevent pregnancy. However, this is not enough. There is a need to communicate or to increase the level of interaction between mothers and adolescents. This closeness will allow adolescent women to share their concerns, contributing to making them aware of the risks involved in sexual relations, and the importance of postponing sex or having protected sex, as shown by other studies. For example, DiClemente et al. (2001) found that less frequent parent–adolescent communication was associated with adolescents' non-use of contraception, including condoms, and lower self-efficacy to negotiate safer sex. Parent–child sexual communication can significantly delay sexual initiation and increases the likelihood of safe sexual practices (Maina et al. 2020). Once again, intervention in the community to induce communication surrounding sexuality between adult women and adolescents and Comprehensive Sexuality Education appear to be necessary. These two proposed actions will provide information about sexuality, not only for young women but also for adult women. Parental involvement in sexual health education can increase the reach and timeliness of this education, providing more skills-based learning, and resulting in improved parent-child communication overall (Saskatchewan Prevention Institute 2017). For young women, comprehensive sexuality education is an opportunity to develop and understand their values and attitudes, as well as to develop the interpersonal skills necessary for making safe decisions despite social pressures (Saskatchewan Prevention Institute 2017).

Regarding beliefs around contraception, although participants understand the need to inform their daughters about precautions for a healthy sexual and reproductive life, there is no consensus about the use of modern contraception by adolescents, since it is perceived as causing infertility (Capurchande et al. 2016). For some participants, there is no need to use contraception before experiencing pregnancy, since there is no proof

that this adolescent can procreate. This is consistent with several studies (Mariano 2014) revealing how infertile women may suffer discrimination for not being able to bear a child. In southern Mozambique, where ancestral descent is patrilineal, the majority of the population marries in a traditional way termed *lobolo*; in some extreme situations, the family of an infertile woman have to reimburse the male partner's family the *lobolo* that was paid for her (Mariano 2014; Hollos and Whitehouse 2014). The participant position regarding contraception methods appears to have a dual and contradictory concept. In that position, beliefs are embedded, which might be a call for bringing people together to discuss and exchange views. For most participants, particularly in Quelimane, all pregnancies have to end in birth because procreation is important and meaningful for their experience. In this sense, as Kumar (2018) highlighted, induced abortion challenges social norms that insist women must always be mothers to be "real" women. Further, abortion is linked to non-procreative sex, challenging traditional ideas of female sexuality, gender roles, and women's agency, which is corroborated by Makleff et al. (2019). Although there were some differences between the two study sites, the causes of the decision to terminate a pregnancy were at intrapersonal, interpersonal, cultural, and environmental levels. Intrapersonal causes were only mentioned in Maputo city, where the circumstances in which pregnancy occurred led to the perception of a lack of interest in procreation. This perception may be linked to the idea and meaning of a child that is given in Maputo city. A child does not, perhaps, have as much weight in terms of meaning for the parent's future as it is given in Quelimane city; this was also found by Arnfred (2011) in northern Mozambique. In Quelimane city, religious factors combined with the social organization of kinship, which is matrilineal, strongly influence values attributed to the process of giving birth, understood as a sign, and the natural process of womanhood, and strongly influence these values much more than in Maputo city. All pregnancies are socially welcomed, since having a child is perceived as synonymous with family enrichment. In contrast, in Maputo city, maybe because of it being a big city with mixed cultures and high cost of life, procreation is not a priority, in particular for adolescents enrolled at school or integrated into the labour market. Thus, in Maputo, abortion seems like a solution for the fulfilment of their dreams (Gbagbo et al. 2015; Loi et al. 2018).

## 5. Conclusions

This study allowed us to understand that intergenerational conflicts result in mothers missing out on the opportunity to transfer knowledge about sexual and reproductive health to their daughters. Also, mothers perceive that available practices concerning early pregnancy and attitudes on decision making regarding how and when to terminate a pregnancy among adolescents are linked to a lack of respect by young women and to a weak dialogue on sexuality. Adult women responsible for imparting knowledge on reproduction and sexuality are no longer playing power control in defining who and when adolescent women should be counselled, resulting in adolescent womens' risky behaviour as well as in the weak relationship between adults and adolescent women. Strategies to improve relationships between adults and adolescent women are necessary, since positive relationships can reduce girls' risky sexual behaviour (Schwandt and Underwood 2013). By improving relationships between these two groups of women, the flow of knowledge will increase, and adolescent women will be empowered to negotiate the practice of safe sex, which in turn can contribute to the reduction of unwanted pregnancy and the need to induce abortion and ensure a healthy transition from childhood to adulthood.

**Author Contributions:** M.F. participated in all process of study. Conceptualization: design of study, data collection, analysis, interpretation, and writing of the initial manuscript. K.M., C.A. and P.D. participated in design, analysis and manuscript editing. R.C. participate in data analysis and interpretation. All authors have read and agreed to the published version of the manuscript.

**Funding:** This research was supported by VLIR-UOS through cooperation between Ghent University, Belgium and Universidade Eduardo Mondlane, Mozambique.

**Institutional Review Board Statement:** The study was conducted in accordance with the Declaration of Helsinki, and Ethical approval was obtained from the Institutional Committee of the Faculty of Medicine of Eduardo Mondlane University and the National Bioethical Committee for the health of the Ministry of Health (IRB00002657), and the ethical committee of the Ghent University Hospital (PA 2015/043).

**Informed Consent Statement:** Informed consent was obtained from all subjects involved in the study.

**Data Availability Statement:** Dataset is part of a broader research for the fulfilment of the requirements for the degree of Doctor of Philosophy of the first author. It may be shared, if necessary, on reasonable request.

**Conflicts of Interest:** The authors declare no conflict of interest.

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
