# Peer review of "Adults’ Perceptions on Adolescent Attitudes towards Pregnancy and Abortion in Maputo and Quelimane Cities, Mozambique: An Exploratory Qualitative Study"

_socsci, doi:10.3390/socsci12010029_

Round 1
Reviewer 1 Report
I have enjoyed reading this paper about older women’s understanding of pregnancy and abortion in two areas of Mozambique. I think the paper carries an important insight, however, I am recommending a revise & resubmit before considering the paper for publication in Social Sciences. I will detail my points here below:
The paper lacks a literature review/theoretical contribution section. For this reason, it is unclear what scholarship it is contributing to. Some of the references used in the Introduction are decidedly dated (Gage, 1998 for example), and therefore do not enable the insight generated here to be woven within more current understanding. I recommend the author carries out a novel search to identify literature that would offer the much needed background. The work of Groes-Green and Manuel, among others, appear as necessary.
In line with the above, the authors need to clarify what the main argument/s of the paper are. What is the message that the data lead on to? That also needs to come out much earlier. I would encourage the author to add a section that clearly signposts how the paper develops, alongside its main arguments.
I have found a significant amount of typos throughout the paper, I recommend careful proofreading before submitting. Also, why is the word pregnancy always capitalised?
p. 2: why does the structure of society (patri/matrillinearity) matters in this study? There is a clear line of inquiry here, but it is somehow left out of the flow.
Methodology: I would like to know more about the FGs: were they structured, semi-structured? How were they managed and recorded? Who did the transcription and how was quality ensured? Had participants been pregnant in the past and sought an abortion themselves? Were they asked to comment on perceived trends in their communities?
I would like some clarification on the following: participants are older women, who have been asked to comment on teenage pregnancy – I am not clear on whether they are commenting on it because they have experienced it themselves? If not, their insight remains valuable as contextual and cultural, but I would want this framed more clearly.
Ethics: thanks for providing the references, it would be great to know a bit more about what the areas of care were, and whether the researcher has specific support (aftercare? NGO? Counselling?) in the generation of the dataset or after.
The paper relies on qualitative methodology. However, the structure of the paper does not align with the methodology. For example, the tables and the order of data in them is hard to weave in within a FG approach. Strangely, an ecological frame is really not adopted, as data are presented as disconnected from their context and somehow reified within the eyes of the author. I would suggest a more realistic presentation of findings that are aligned with the context in which they were proffered, so that the analysis becomes clearer and more compelling.
Really odd formatting that makes is hard to engage with the flow. Some quotes are in italic, some are not. Does this mean there is a difference between quotes?
The discussion section distinct from data is problematic: this structure fits well with a quantitative approach, but undersells the data and the analysis from a qualitative methodology. I would advise that the author rethinks the structure of the paper to offer a thematic overview of their findings, which constructs the argument/s the paper proposes.
The paper identifies a disconnect between young people and older generation, which is assumed as leading to lack of information regarding sexuality and fertility. However, the study did not yield any information regarding where and how young people develop knowledge regarding sexuality. The fact they do not seek knowledge from older generation does not mean they have no knowledge. Where and how do they develop it then?
The discussion is structured around 2 topics: 1) Inter-generational conflicts and reduce the importance paid to traditional knowledge transfer [check grammar], 2) Beliefs about contraception and the importance of procreation in the Mozambican culture. I would like the author to clarify how the dataset led to the development of these two topics.
I have found some of the language used in this article constructs young people as passive, and the educational journey as ‘transfer’. For example in page 11 the author talks about ‘transference of knowledge’ or ‘instruct’. This view is quite limited and fails to consider young people’s voices.
I think this last is the most problematic aspect: the paper purports to be about young people, but does not actually involve them as participant. I would suggest the author rephrases the article around different aims and objective, which would frame the dataset more appropriately.
Author Response
Thank you for the critical comments. it was a good challenge.

Reviewer 2 Report
Dear authors,
First of all, I want to congratulate you for your work "Female adults' perceptions on adolescent pregnancy and abortion in Maputo and Quelimane cities, Mozambique: An exploratory qualitative study", deals with a topic that, although it has been highly explored in the literature, is really relevant. In general, the work is constructed in a correct way, however, I make a series of recommendations that, from my point of view, can help improve the understanding of your article.
In the first place, regarding questions of form, I would recommend a review of the format and style of your manuscript, it is also important that during this review the citation rules of the journal be carried out, especially reviewing the links provided, many of which are not in operation.
Second, regarding content issues, the following issues are recommended:
Introduction:
As they point out: “The objective of this study is to explore the perceptions of adult women (a) on causes 63 and solutions for adolescent pregnancy (b) on the decision-making processes of abortion; 64 and, (c) on the new abortion law. The study is a contribution to a better understanding of 65 the decision-making processes among adolescent women in Mozambique.”, however, no type of theoretical approximation is made to the existing antecedents in the field. Specifically to possible studies that explore these perceptions and can later help us maintain a theory-based discussion. It is recommended that you explore this.
Methodology:
It is recommended that the thematic analysis procedure be explored further, preferably by exploring the different categories/codes that emerged and the relationship with the content of the interviews. This can contribute to greater transparency regarding the investigation procedure.
In limitations they state that they do not intend to generalize the results. Could this be explored in methodology considering the results as mid-range theories? Perhaps it would be convenient to explore this, although I leave this point as an open question.
Results:
In relation to the previous section, perhaps it would be interesting to codify the participants individually, in such a way that, in the indicated fragments, as well as in the categories that are exposed, it is possible to see graphically who we are talking about, or how representative they are. are these categories.
Discussion:
It would be interesting if it were linked to the theoretical framework (absent at the moment) and it is recommended to update some of the references.
My best wishes,
Reviewer.
Author Response
thank you for your comments. we learned more.

Round 2
Reviewer 2 Report
My concerns have been take into account.
Author Response
Social Sciences Journal
26 December 2022
Subject: Revision of manuscript “Adults’ perceptions on adolescent attitudes towards pregnancy and abortion in Maputo and Quelimane cities, Mozambique: An exploratory qualitative study” (adapted title)
Your reference: socsci-1907175
Dear Dr. Ariana Guga, Assistant Editor of the Social Sciences journal
Thank you for a careful revision of our manuscript entitled “Adults’ perceptions on adolescent attitudes towards pregnancy and abortion in Maputo and Quelimane cities, Mozambique: An exploratory qualitative study”. In the following pages are our point-by-point responses to each of the comments of the editor. Attached, you can find the revised version of our manuscript. All changes and information added to the manuscript are in red.
We would like to thank you and look forward to having the article published in your journal.
On behalf of the writing team.
Kind regards,
Mónica Frederico
The following minor revisions are required to get the paper to publication standard:
(1) The title refers to ‘Adults influences on adolescent attitudes …’ while study objectives refer to ‘…perceptions of adult women …’ . The title could be better aligned with study objectives and methods which focus on ‘adults’ perception on adolescents’ pregnancies and induced abortion decision making’.
R: Thank you for this observation. We made changes in the title to align with the objective of the manuscript. Now it can be read: Explore adults’ perceptions on adolescent attitudes towards pregnancy and abortion in Maputo and Quelimane cities. See amendments in lines 5 and 6 of the abstract and introduction section, Pg 2, lines 89 and 90.
(2) Thorough proof reading is necessary to correct for typo-grammatical errors. These include, but not limited to:
- ‘Data was analysed …’ should be ‘Data were analysed…’ – in abstract ;
R: Done. Now in the abstract section, line 8, the word was, was replaced by were. We have also proof Read the manuscript and corrected all typo-grammatical errors.
- Final statement of page 2 not clear.
R: Noted with thanks. We decided to delete it from the manuscript. Indeed, we have forgotten to remove it.
- 2nd paragraph of section 2.2 – ‘two researchers were involved e this FGDs, …’ – check/correct statement.
R: Correction was made. The sentence reads as follows: Two researchers were involved in these FGDs. It was inserted into the manuscript, line 134.
- The result section could be better formatted
R: Thank you for this observation. We improved the format of the result section by changing the presentation of all quotes in the manuscript. See the whole section revised.